# Closed Formula for Transport across Constrictions

**DOI:** 10.3390/e25030470

**Published:** 2023-03-08

**Authors:** Paolo Malgaretti, Jens Harting

**Affiliations:** 1Helmholtz Institute Erlangen-Nürnberg for Renewable Energy, Forschungszentrum Jülich, 90429 Erlangen, Germany; 2Department of Chemical and Biological Engineering and Department of Physics, Friedrich-Alexander-Universität Erlangen-Nürnberg, 90429 Erlangen, Germany

**Keywords:** porous materials, entropic barrier, transport

## Abstract

In the last decade, the Fick–Jacobs approximation has been exploited to capture transport across constrictions. Here, we review the derivation of the Fick–Jacobs equation with particular emphasis on its linear response regime. We show that, for fore-aft symmetric channels, the flux of noninteracting systems is fully captured by its linear response regime. For this case, we derive a very simple formula that captures the correct trends and can be exploited as a simple tool to design experiments or simulations. Lastly, we show that higher-order corrections in the flux may appear for nonsymmetric channels.

## 1. Introduction

It is common to experience long queues when a constriction occurs on a highway [1,2]. Such an (unlucky) phenomenon is clearly the result of “local” confinement: due to constriction, vehicles slow down, hence reducing the local “mass” flux as compared to the clear part of the highway. Such a local reduction in mass flow causes the onset of annoying queues that we sometimes experience. This phenomenon does not only occur on highways. It becomes a major issue close to emergency exits in the case of panic [3]. The very same dynamics also occurs at smaller scales and for simpler systems. For example, it is a common experience that it is difficult to extract pills from a container if the opening is too small. Pills tend to “clog”, i.e., to form stable structures close to the opening of the container that prevent pills from going out. Very similar dynamics occurs in silos containing crops [4], erosion [5], suspensions of hard and soft particles [6,7,8], herds of sheep [9], and in the onset of panic in ants [10] and even humans [11].

The effect of confinement does not have to be unpleasant, as it is for traffic jams, or inconvenient, as it is for the clogging of silos. Tuning the shape of the confining media can also be an intriguing and novel way to control the dynamics of the confined system. For example, microfluidic devices exploit variations in the section of the microchannels of which they are composed to control fluid dynamics and induce the formation of droplets [12,13,14,15]. Similarly, tuneable resistive pulse sensing (TRPS) techniques exploit micro- and nanopores to analyze small particles ranging from a few tens of nanometers up to the micrometric scale [16]. In particular, TRPS was used to directly detect antibody–antigen binding [17], to measure the electrophoretic mobility of colloidal particles [18], to perform single-molecule detection [19], and to measure the zeta potential of nanometric particles [20]. Chromatographic techniques were also developed to separate micro- or nanoparticles depending on their size and surface properties [21,22,23,24]. Lastly, at even smaller scales, nanopores were designed to sequence DNA molecules [25].

Transport in confinement is not only relevant for particle detection/analysis. Indeed, the flow of fluids across a porous medium is crucial in diverse scenarios. For example, oil recovery industries put much effort into developing techniques to maximize the extraction of oil from the rock matrix in which it is embedded [26,27]. Similarly, understanding the dependence of the flow of water on the porosity of the soil is crucial in environmental sciences [28]. Moreover, diverse technologies related to the energy transition such as blue energy [29], hydrogen technology [30,31], electrolyzers and fuel cells [32,33], and CO_2_ segregation [34] rely on the transport of (charged) chemical species across nanoporous materials.

Lastly, several biological systems are controlled by the transport of confined complex fluids. For example, neuronal transmission relies on the transport of neuroreceptors among neurons and to their specific binding sites [35]. Moreover, cell regulation relies on the proper tuning of the concentrations of electrolytes inside the cell. Such a regulation occurs via dedicated pores and channels whose shape renders them very sensitive to specific ions [36,37,38,39,40], and RNA is transported across the nuclear membrane [41,42,43]. Moreover, the lymphatic and circulatory systems in mammals rely on the transport of quite heterogeneous suspensions composed of a variety of components, spanning from the nanometric size of ions up to the micrometric size of red blood cells across varying-section elastic pipes [7,44,45,46]. Lastly, the survival of plants relies, at large scales, on the proper circulation of liquid (sap) along the trunk [47] and, at short scales, on the cytoplasmic streaming within the cells [48].

All the above-mentioned systems rely on the dynamics under confinement. Therefore, understanding the dynamics and transport properties of confined complex systems such as ions, molecules, polymers, colloidal particles, and suspensions is of primary importance for the understanding of a wide spectrum of phenomena and for the development of technological applications. Identifying the relevant parameters controlling key features such as transport or phase transitions opens a new route for controlling the dynamics of confined systems upon tuning the geometry of the confining media.

There has been no systematic study of the dependence of the dynamics of confined systems upon changing the shape of the confining walls. The main reason is the large effort that such a study requires. Indeed, *experimentally* tuning the shape of a pore is a tremendous task since, if possible at all, it requires to synthesize a new item from scratch every time. On the *theoretical* side, studying the dynamics and the transport of confined systems is a tremendous task since it requires to capture several length, time, and energy scales. In fact, the length scales range from the nanometric, typical for ions and van der Waals interactions, to the micrometric, of colloids, polymers, and macromolecules up to the mm/cm scale of microfluidic devices. Concerning time scales, the spectrum spans the diffusion time of small particles and ions over their size ∼μs up to the long time scales typical of transport ∼s. Concerning energy scales, they range from thermal energy $k_{B}T$ ($∼10^{-21}\;J$) up to van der Waals and electrostatic interactions whose magnitude can be of several $k_{B}T$. On the top of these “direct” interactions, the effective interactions induced by the confinement should also be accounted for. For example, squeezing a deformable object such as a polymer or a vesicle through a constriction can require quite an amount of energy that can easily reach the order of 100–1000$\;k_{B}T$. Given such complexity, one typically relies on numerical techniques such as molecular dynamics. However, the wide range of interactions (van der Walls, electrostatic, etc.)and the wide range of time and length scales impose advancing numerical approaches capable of properly resolving the smallest length, time, and energy scales. At the same time, such an approach should also resolve the large length, time, and energy scales. Accordingly, the numerical route becomes quite demanding from the perspective of computational time.

Since the experimental and the numerical route are quite expensive, an approximated analytical route based on some controllable expansions may become appealing. Intriguingly, it is possible to obtain simple analytical models that capture some features of the dynamics of confined systems. The key idea is to “project” the dynamics of the system onto some relevant coordinate (in chemistry, sometimes called a “reaction coordinate”) and then to study the dynamics of these few (typically one) degrees of freedom. For example, in the case of polymer translocation across pores, the most important observable aspect is the time that the polymer takes to cross from one side of the pore to the other. Therefore, the relevant degree of freedom is the position of the center of mass of the polymer, whereas the degrees of freedom associated with the position of the monomers can be integrated out.

In this contribution, we briefly review the derivation of the Fick–Jacobs approximation [49,50,51,52,53,54,55,56] and its use in studying transport across corrugated pores and channels. The Fick–Jacobs approximation is applicable to the transport of ions [57,58,59,60,61], colloids [62,63,64,65,66], rods [67], polymers [68,69,70], and, more recently, even chemical reactors [71] and pattern-forming systems [72]. The validity of the Fick–Jacobs approximation was numerically assessed in the case of polymers [68], for infinitely diluted [73] and denser [66,74] (up to 40% volume fraction) colloidal suspensions, and detailed simulations of electrolytes in varying-section channels [60] qualitatively reproduced the analytical predictions [57]. Lastly, the role of hydrodynamic interactions was addressed in the case of rods diffusing across a varying-section channel [75]. However, these experimental results are quite compatible with a simple model that does indeed disregard hydrodynamic interactions [69]. So, for the above-mentioned case, hydrodynamic interactions seem to mildly affect the Fick–Jacobs approximation. In contrast, the dynamics of confined active systems is very sensitive to their effective hydrodynamic interactions with the channel walls [76,77,78,79].

In the following, we rederive the Fick–Jacobs approximation with particular emphasis on the regime in which the current is proportional to the applied force. In such a regime, it is possible to derive a closed formula that accounts for the dependence of the flux on the geometry of the channel. Interestingly, our derivation naturally highlights a few relations between the underlying Smoluchowski equation and the linear response theory. Even though this work was motivated by the transport in confined pores and channels, the results we derived are valid for all 1D systems (independently of the physical origin of the effective potential) in the dilute regime (for which mutual interactions can be neglected) and whose dynamics is governed by the Smoluchowski equation (i.e., in the overdamped regime).

## 2. Model

In the following, we are interested in the transport of a single colloidal particle confined in an axially symmetric channel characterized by its half section (see Figure 1 for a sketch of the system):(1)$$\begin{array}{c} h\left(x\right)=h_{0}+h_{1}\cos\left(2\pi \frac{x}{L}\right)\;. \end{array}$$
and period *L*. In the case of axis-symmetric channels, in cylindrical coordinates, the time evolution of probability density ρ is governed by the Smoluchowski equation:(2)$$\begin{array}{c} \dot{\rho }\left(x,r,t\right)=\nabla \cdot \left[D\nabla \rho (x,r,t)+D\beta \rho (x,r,t)\nabla W(x,r)\right]\;, \end{array}$$
where *D* is the diffusion coefficient, $\beta^{-1}=k_{B}T$ is the inverse thermal energy, $k_{B}$ the Boltzmann constant, *T* the absolute temperature and
(3)$$\begin{array}{c} W\left(x,r\right)=\left\{\begin{array}{cc} \phi (r) & r<h(x)\\ \infty & \mathrm{else} \end{array}\right. \end{array}$$
is the effective potential responsible for both confining the particle within the channel and for additional soft interactions, ϕ(r) with the channel walls. For smoothly varying channel cross-sections $\partial_{x}h\left(x\right)\ll 1$, it is possible to factorize the probability density [49,50,53,54,55,56]
(4)$$\begin{array}{c} \rho \left(x,r,t\right)=p\left(x,t\right)\frac{e^{-\beta W(x,r)}}{e^{-\beta A(x)}}\;, \end{array}$$
where
(5)$$\begin{array}{c} A\left(x\right)=-k_{B}T\ln\left[\frac{1}{\pi h_{0}^{2}}\int_{0}^{\infty }e^{-\beta W(x,r)}rdr\right] \end{array}$$
is the local free energy [59]. Moreover, integrating along the radial direction leads to
(6)$$\begin{array}{c} \dot{p}\left(x,t\right)=\partial_{x}\left[D\partial_{x}p\left(x,t\right)+D\beta p\left(x,t\right)\partial_{x}A\left(x\right)\right]\;. \end{array}$$

Such a procedure is called *Fick–Jacobs approximation* [49,50,56]. Its regime of validity was assessed by several groups [51,52,54,62,73,80,81,82,83,84]. In particular, the quantitative reliability of the Fick–Jacobs approximation can be enhanced by introducing a position-dependent diffusion coefficient [51,52,54,62,73,80,81,82,83,84], D(x), hence leading to the following set of equations:(7)$$\begin{array}{cc} \dot{p}\left(x,t\right) & =-\partial_{x}J\left(x,t\right) \end{array}$$(8)$$\begin{array}{cc} \frac{J}{D(x)} & =-\partial_{x}p\left(x\right)-\beta p\left(x\right)\partial_{x}A\left(x\right)\;. \end{array}$$Equation (Equation 8) is completed with the following boundary conditions: (9)$$\begin{array}{cc} p(-L) & =p(L) \end{array}$$(10)$$\begin{array}{cc} \int_{-L}^{L}p\left(x\right)dx & =1. \end{array}$$We decomposed effective force $-\partial_{x}A\left(x\right)$ as the net force:(11)$$\begin{array}{c} f=-\frac{1}{2L}\int_{-L}^{L}\partial_{x}A\left(x\right)dx=-\frac{\Delta A}{2L} \end{array}$$
and
(12)$$\begin{array}{c} A_{eq}\left(x\right)=A\left(x\right)+fx. \end{array}$$
where *f* is the net force responsible of the flux, and $A_{eq}\left(x\right)$ is all the other conservative forces that do not give rise to any flux. In the following, we expand both flux *J* and density *p* in the equilibrium case:(13)$$\begin{array}{cc} J= & J_{0}+J_{1}+J_{2}+\ldots \end{array}$$(14)$$\begin{array}{cc} p(x)= & p_{0}\left(x\right)+p_{1}\left(x\right)+p_{2}\left(x\right)+\ldots \end{array}$$Due to Equation (Equation 10), at the zero-th order, we have
(15)$$\begin{array}{cc} \int_{-L}^{L}p_{0}\left(x\right)dx & =1\;. \end{array}$$This implies
(16)$$\begin{array}{cc} \int_{-L}^{L}p_{n}\left(x\right)dx & =0\;\;\;\;\forall n\neq 0 \end{array}$$

Accordingly, at order zero, we have
(17)$$\begin{array}{cc} p_{0}\left(x\right) & =\tilde{p}e^{-\beta A_{eq}\left(x\right)} \end{array}$$
(18)$$\begin{array}{cc} J_{0} & =0 \end{array}$$
(19)$$\begin{array}{cc} \tilde{p} & =\frac{1}{\int_{-L}^{L}e^{-\beta A_{eq}\left(x\right)}dx}\;. \end{array}$$At the generic *n*-th order, we have
(20)$$\begin{array}{c} \frac{J_{n}}{D(x)}=-\partial_{x}p_{n}\left(x\right)-\beta p_{n}\left(x\right)\partial_{x}A_{eq}\left(x\right)+\beta p_{n-1}\left(x\right)f\;, \end{array}$$
the solution of which reads
(21)$$\begin{array}{c} p_{n}\left(x\right)=e^{-\beta A_{eq}\left(x\right)}\left[\int_{-L}^{x}\left[\beta p_{n-1}\left(y\right)f-\frac{J_{n}}{D(y)}\right]e^{\beta A_{eq}\left(y\right)}dy+\Pi_{n}\right]\;. \end{array}$$Here, $J_{n}$ and $\Pi_{n}$ are integration constants. Imposing periodic boundary conditions $p_{n}\left(-L\right)=p_{n}\left(L\right)$ and recalling that $A_{eq}\left(-L\right)=A_{eq}\left(L\right)$ leads to
(22)$$\begin{array}{c} \int_{-L}^{L}\left(\frac{J_{n}}{D(y)}-\beta p_{n-1}\left(y\right)f\right)e^{\beta A_{eq}\left(y\right)}dy=0\;, \end{array}$$
with
(23)$$\begin{array}{c} J_{n}=\beta f\frac{\int_{-L}^{L}p_{n-1}\left(y\right)e^{\beta A_{eq}\left(y\right)}dy}{\int_{-L}^{L}\frac{e^{\beta A_{eq}\left(y\right)}}{D(y)}dy}=\beta f\tilde{p}\frac{\int_{-L}^{L}\frac{p_{n-1}\left(y\right)}{p_{0}\left(y\right)}dy}{\int_{-L}^{L}\frac{e^{\beta A_{eq}\left(y\right)}}{D(y)}dy}\;. \end{array}$$
In the last step, we used Equation (Equation 17). Lastly, $\Pi_{n}$ is determined by imposing Equations (Equation 15) and (Equation 16):(24)$$\begin{array}{c} \Pi_{n}=-\tilde{p}\int_{-L}^{L}e^{-\beta A_{eq}\left(x\right)}\int_{-L}^{x}\left[\beta p_{n-1}\left(y\right)f-\frac{J_{n}}{D(y)}\right]e^{\beta A_{eq}\left(y\right)}dydx\;. \end{array}$$
At the leading order in the force, Equations (Equation 21) and (Equation 23) read: (25)$$\begin{array}{cc} p_{1}\left(x\right)= & \;e^{-\beta A_{eq}\left(x\right)}\left[\beta f\tilde{p}\left(x+L\right)-J_{1}\int_{-L}^{x}\frac{e^{\beta A_{eq}\left(y\right)}}{D(y)}dy\right]\;, \end{array}$$(26)$$\begin{array}{cc} J_{1}= & \;\frac{2\beta fL}{\int_{-L}^{L}e^{-\beta A_{eq}\left(x\right)}dx\int_{-L}^{L}\frac{e^{\beta A_{eq}\left(x\right)}}{D(x)}dx}\;. \end{array}$$
Interestingly, from Equation (Equation 26), it is possible to identify a force-independent channel permeability:(27)$$\begin{array}{c} \chi =\frac{2\beta L}{\int_{-L}^{L}e^{-\beta A_{eq}\left(x\right)}dx\int_{-L}^{L}\frac{e^{\beta A_{eq}\left(x\right)}}{D(x)}dx}\;. \end{array}$$
As expected, Equation (Equation 27) agreed with the derivation of the effective diffusion coefficient for a particle at equilibrium and in the presence of entropic barriers [85,86]. This is in agreement with the linear response theory, within which the transport coefficients that determine the flux under external forces can be determined from equilibrium properties. In the case in which the density at the ends of the channel differs (instead of the periodic boundary conditions considered here), the Fick–Jacobs approximation agrees with closed formulas that do not rely on the smooth variation of the channel [58,87].

Some general remarks can be derived in the case of fore-aft symmetric channels, for which $A_{eq}\left(x\right)=A_{eq}\left(-x\right)$, and diffusivities, D(x)=D(−x). For such cases, the magnitude of the flux should depend solely on the magnitude of the force and not on its sign. This implies that
(28)$$\begin{array}{c} J_{2n}=0,\;\forall n>0\;. \end{array}$$
In order to proceed, for fore-aft symmetric f(x) and g(x), the following equality holds:(29)$$\begin{array}{c} \int_{-L}^{L}g\left(x\right)\int_{-L}^{x}f\left(y\right)dydx=\frac{1}{2}\int_{-L}^{L}f\left(x\right)dx\int_{-L}^{L}g\left(x\right)dx\; \end{array}$$
Enforcing the condition in Equation (Equation 28) into Equation (Equation 23), and using the last expression leads to
(30)$$\begin{array}{c} \Pi_{n}=0,\;\forall n>0\;. \end{array}$$
Substituting again into Equation (Equation 23) eventually leads to
(31)$$\begin{array}{c} J_{n}=0,\;\forall n\geq 1\;. \end{array}$$
Even though $\Pi_{n>0}=0$ and $J_{n>1}=0$, the density profile was still sensitive to higher-order corrections in the force, i.e., in general, $p_{n}\neq 0$. According to this analysis, Equation (Equation 26) was not just the linear contributions to the flux; rather, it also provided the exact expressions at every order in the external force. The outcome of this analysis is intuitive since it states that, for noninteracting systems confined within fore-aft symmetric channels, nonlinear effects are absent. The same results are valid for any 1D problem with such a symmetry.

In contrast, if neither potential A(x) nor diffusion profile D(x) have a defined parity, then the left-right symmetry is broken, Equation (Equation 28) does not hold anymore, and a diode effect may set for sufficiently large external forces. We could assess the dependence of the diode effect on the geometry of the channel by calculating the following:(32)$$\begin{array}{c} J_{2}=\beta f\frac{\int_{-L}^{L}\int_{-L}^{x}\;\;\;\beta \tilde{p}f-\frac{J_{1}}{D(y)}e^{\beta A_{eq}\left(y\right)}dy+\Pi_{1}dx}{\int_{-L}^{L}\frac{e^{\beta A_{eq}\left(y\right)}}{D(y)}dy}\;. \end{array}$$Using
(33)$$\begin{array}{c} \Gamma \left(x\right)=\int_{-L}^{x}\frac{e^{\beta A_{eq}\left(y\right)}}{D(y)}dy \end{array}$$
and the definition of $J_{1}$, we obtain
(34)$$\begin{array}{c} J_{2}=\frac{\beta f}{\Gamma (L)}\int_{-L}^{L}\beta \tilde{p}f\left(x+L\right)-2\beta \tilde{p}fL\frac{\Gamma (x)}{\Gamma (L)}+\Pi_{1}dx\;. \end{array}$$Lastly, using the definition of $\Pi_{1}$, we obtain
(35)$$\begin{array}{cc} J_{2}= & \frac{{\left(\beta fL\right)}^{2}\tilde{p}}{\Gamma (L)}\frac{1}{L}\int_{-L}^{L}\left[\left(\frac{x}{L}+1\right)-2\frac{\Gamma (x)}{\Gamma (L)}\right]\left[1-e^{-\beta A_{eq}\left(x\right)}\right]dx\;. \end{array}$$

### 2.1. Transport across Free Energy Barriers

In the case of the transport of pointlike particles across 3D varying-section channels with axial symmetry, the effective potential reads:(36)$$\begin{array}{c} A_{eq}^{\left(id\right)}\left(x\right)=-2k_{B}T\ln\left[\frac{h(x)}{h_{0}}\right]\;, \end{array}$$
where h(x) is the local half-section of the channel, and $h_{0}$ its average value (see Figure 1). Accordingly, Equation (Equation 26) reads
(37)$$\begin{array}{c} J_{id}=\frac{2\beta fL}{\int_{-L}^{L}\frac{h^{2}\left(x\right)}{h_{0}^{2}}dx\int_{-L}^{L}\frac{h_{0}^{2}}{h^{2}\left(x\right)D\left(x\right)}dx}\;. \end{array}$$In the case of micro- or nanoparticles that undergo solely excluded volume interactions with the channel walls, the effective channel half-section becomes h(x)−R, where *R* is the particle size, and we obtain
(38)$$\begin{array}{c} A_{eq}^{\left(pcl\right)}\left(x\right)=-2k_{B}T\ln\left[\frac{h(x)-R}{h_{0}}\right]\;, \end{array}$$
which leads to
(39)$$\begin{array}{c} J_{pcl}=\frac{2\beta fL}{\int_{-L}^{L}\frac{{\left(h\left(x\right)-R\right)}^{2}}{h_{0}^{2}}dx\int_{-L}^{L}\frac{h_{0}^{2}}{{\left(h\left(x\right)-R\right)}^{2}D\left(x\right)}dx}\;. \end{array}$$
$R<h_{0}-h_{1}$ for the particle to be able to cross the channel. Lastly, several groups showed that the Fick–Jacobs approximation can be improved by assuming a position-dependent diffusion coefficient [49,50,53,54,81,82,83,84]. There is general agreement that the approximated formula for the diffusion coefficient reads [50] (or is, in practice, equivalent to):(40)$$\begin{array}{c} D\left(x\right)=\frac{D_{0}}{\sqrt{1+{\left(\partial_{x}h\left(x\right)\right)}^{2}}}\;. \end{array}$$

### 2.2. Piecewise Linear Potential and Homogeneous Diffusion Coefficient

For analytical insight, it can be useful to approximate effective potential A(x) with
(41)$$\begin{array}{c} A_{eq}\left(x\right)=-\frac{\Delta A_{eq}}{L}\left|x\right|\;, \end{array}$$
where
(42)$$\begin{array}{c} \Delta A_{eq}=A_{eq}^{max}-A_{eq}^{min} \end{array}$$
is the piecewise linear difference between the maximal and minimal values of $A_{eq}$. Moreover, if we assumed that the diffusion coefficient was homogeneous:(43)$$\begin{array}{c} D\left(x\right)=D_{0} \end{array}$$
we obtained
(44)$$\begin{array}{cc} \int_{-L}^{L}e^{\beta A_{eq}\left(x\right)}dx= & \frac{2L}{\beta \Delta A_{eq}}\left(1-e^{-\beta \Delta A_{eq}}\right) \end{array}$$
(45)$$\begin{array}{cc} \int_{-L}^{L}e^{-\beta A_{eq}\left(x\right)}dx= & \frac{2L}{\beta \Delta A_{eq}}\left(e^{\beta \Delta A_{eq}}-1\right) \end{array}$$
Lastly, by substituting the last expressions into Equation (Equation 27), we obtained an approximated expression for the following permeability:(46)$$\begin{array}{c} \tilde{\chi }=\frac{D\beta }{4L}\frac{{\left(\beta \Delta A_{eq}\right)}^{2}}{\cosh(\beta \Delta A_{eq})-1}\;. \end{array}$$
Interestingly, Equation (Equation 46) shows that χ was an even function of $\Delta A_{eq}$. This implies that the transport was insensitive upon flipping the sign of free energy barrier ΔA. Lastly, Equation (Equation 46) shows that χ decayed exponentially with $\beta \Delta A_{eq}$.

## 3. Discussion

The reliability of the Fick–Jacobs approximation, namely, Equation (Equation 26), was addressed for pointlike particles and showed good quantitative agreement for forces up to βfL≃10 [73]. However, Equation (Equation 26) still required to numerically compute integrals, whereas Equation (Equation 46) provided a direct (yet approximated) dependence of $\tilde{\chi }$ on ΔA. Therefore, it is important to address the reliability of Equation (Equation 46) as compared to the full solution of Equation (Equation 26). Indeed, all the panels of Figure 2 show that, the permeability calculated with the piecewise linear model, Equation (Equation 46), showed some discrepancies as compared to the full expression in Equation (Equation 26). In particular, as shown in Figure 2 for the case under consideration ($h_{0}/L=0.1$), the corrections due to the inhomogeneous diffusion (dashed-dotted lines) were indistinguishable from those with a constant diffusion coefficient (dashed lies), and hence did not improve the approximation. On the other hand, Figure 2 shows that the simple formula in Equation (Equation 46) was sufficient to properly capture the trends and could be used to estimate the transport of colloidal particle across porous media. Interestingly, concerning the magnitude of χ, the bottom panels of Figure 2 show that the channel permeability decreased upon increasing the particle size. Interestingly, the decrease was almost linear for larger corrugations of the channel (larger values of ΔS), whereas for smaller values of the corrugation, it plateaued at smaller values of *R*. Lastly, we discuss the dependence of $\tilde{\chi }$ in βΔA as per Equation (Equation 46). As shown in Figure 3, $\tilde{\chi }$ had a maximum for βΔA=0 and then it decayed exponentially for larger values of βΔA. Interestingly, $\tilde{\chi }$ attained values close to unity up to βΔA≃5, i.e., for a free energy barrier much larger than the thermal energy.

The fact that Equation (Equation 46) depended solely on ΔA also allowed for estimating the transport in situations in which the particles may have had some soft interactions with the walls, such as electrostatic interactions. In that case, the free energy barrier depended not only on the size of the particle and the geometry of the channel, but also on the charge of both the particle and the walls of the channels [58,59]. Moreover, Equation (Equation 46) allowed for predicting the transport of soft or deformable objects, such as proteins or polymers [68,69,88].

## 4. Conclusions

We derived closed formulas for transport within linear response theory and for higher-order corrections. In particular, we showed that, for the case of noninteracting systems confined in fore-aft symmetric channels, the higher-order corrections in the flux and the density were both zero. Hence, for fore-aft symmetric channels, the full expression for the flux was indeed the one obtained within the linear response regime. Accordingly, the channel permeability derived within the linear response, as shown in Equation (Equation 27), was related to the well-known expression of the effective diffusion coefficient reported in the literature [85,86]. Moreover, we showed that, within the linear response, the formula for permeability χ, as shown in Equation (Equation 27), could be further simplified by approximating the local free energy with piecewise linear potential (Equation (Equation 41)) to obtain Equation (Equation 46) whose overall drop was determined with the values of the free energy at the bottleneck and at the waist of the channel. We showed that such an approximation provided the correct trends and was reliable within ≃±50%, as shown in the right-hand panels of Figure 2. This feature is crucial, since Equation (Equation 46) can be easily computed and it is valid for all soft interactions between the particle and the channel walls. 

## Figures and Tables

**Figure 1 entropy-25-00470-f001:**
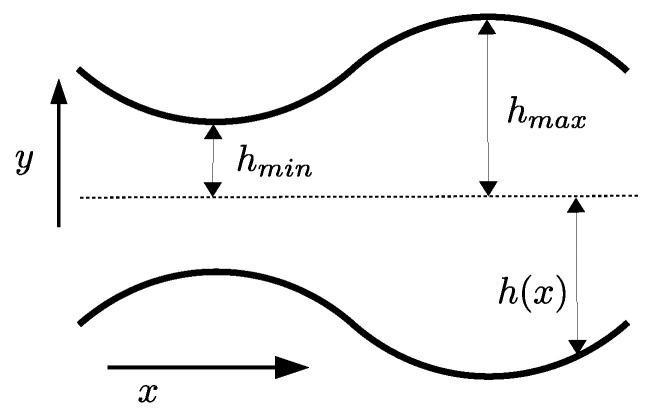
Sketch of a channel with varying-section h(x). The minimal $h_{min}$ and maximal $h_{max}$ amplitudes are marked. The channel was periodic along the *x* direction with period *L*.

**Figure 2 entropy-25-00470-f002:**
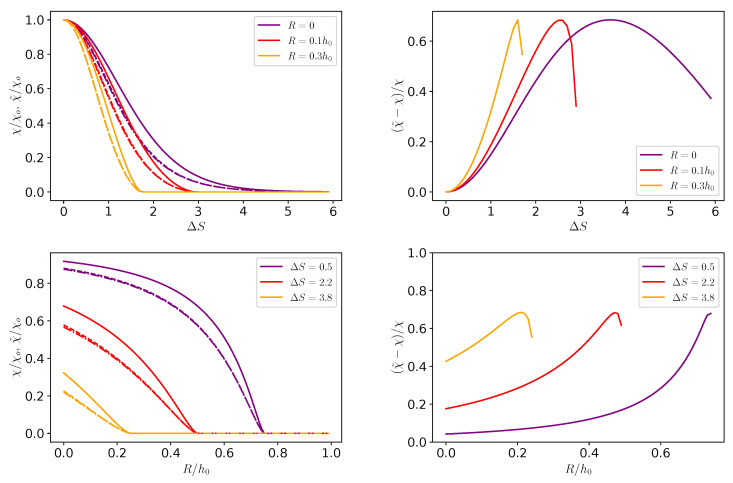
Transport across porous media. (upper left) Permeability χ as obtained form Equation (Equation 46) (solid lines), Equation (Equation 26) with constant diffusion coefficient (dashed lines), and Equation (Equation 26) with a diffusion coefficient as given by Equation (Equation 40) (dashed-dotted lines) normalized by the one across a constant-section channel $\chi_{o}=D\beta /4L$, as a function of the geometry of the channel $\Delta S=\ln\frac{h_{0}+h_{1}}{h_{0}-h_{1}}=\ln\frac{h_{max}}{h_{min}}$ for different values of the particle radius. (upper right) Ratio of $\tilde{\chi }$ over χ normalized by χ for the datasets shown in the left panel. (bottom left) Permeability χ normalized by the one across a constant-section channel $\chi_{o}=D\beta /4L$ as a function of the radius of the particle, *R*, normalized by the average channel width, $h_{0}$, for different channel geometries captured by ΔS. (bottom right) Ratio of $\tilde{\chi }$ over χ normalized by χ for the datasets shown in the left panel.

**Figure 3 entropy-25-00470-f003:**
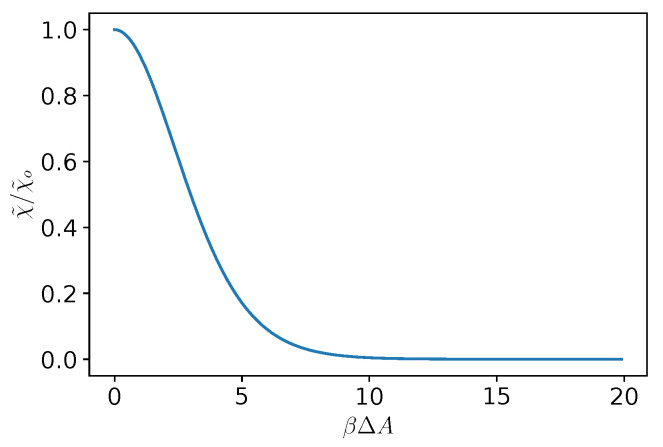
Dependence of approximated channel permeability $\tilde{\chi }$ (as defined in Equation (Equation 46)) normalized by that of a constant section channel $\chi_{o}$ as function of the amplitude of the dimensionless free energy barrier βΔA that encodes the physical properties of the confined system.

## Data Availability

Not applicable.

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
