# Peer review of "Closed Formula for Transport across Constrictions"

_entropy, 2023, doi:10.3390/e25030470_

Round 1

Reviewer 1 Report

 Dear Author:

In the manuscript " Closed formula for the transport across constrictions" authors shown that for point-like Brownian particles confined in symmetric channels, there is no higher order corrections in both the flux and in the density. On the other hand, for symmetric channels the full expression for the flux is the obtained within the linear response regime. Authors also show that within linear response, the formula for the permeability can be further simplified by approximating the local free energy by a piece-wise linear potential.

The results are clear and interesting. But authors must attend some request before been considering for publication.

The following recommendations are intended to make the article readable for potential readers.

Authors do not define variables which appears in equations, at least a couple of times. Not even when they refer to a figure, as in the case of equation (1), where it is never specified who L is. The authors should make a more precise description in the caption of figure 1.

I think it is worth calling the attention of the authors regarding the results recently found in J. Chem. Phys. 156, 071103 (2022), and the articles published subsequently by the same group, regarding the physical interpretation of resistance and permeability.

In general authors must improve the quality of the manuscript before being published.

Author Response

We thank the Reviewer for the very interesting reference mentioned in the report. We have added it in the new version of the manuscript (highlighted) and we have gone through the text in order to improve it. 

Reviewer 2 Report

The manuscript “Closed formula for the transport across constrictions” is interesting and well-written. I consider it deserves publication.

I have just a few very simple remarks:

-        the text in red on page 3 has to be removed and the text corrected accordingly;

-        I think that in Eq. (5), page 4, the integral should be from 0 to infinity, not from -infinity to infinity.

Author Response

We thank the Reviwer for the positive evaluation of our work and for pinpointing these issues that we have corrected for in the new version of the manuscript.

Reviewer 3 Report

The paper by Malgaretti and Harting is an interesting opinion paper focused on deriving simple formulations for transport properties of channel-like ducts, described by means of the Fick-Jacobs formalism. I like the clarity and outline of the paper. The contribution is clear and well written. I recommend publication.

A perhaps naive question to the Authors would be – have there been any MD or rather Brownian Dynamics simulations that would attempt to determine the transport properties of channels with deformed walls? I understand that in this case resolving the interactions, particularly hydrodynamic interactions, is a challenge but perhaps there have been numerical works focusing on addressing various aspects of this problem? I would appreciate if the Authors could comment on this. 

There are a couple of typos I found in the paper. I list them below for convenience. 

– L27 – “micro- and nano-“

– L105 - “underlying”

– remove spurious remarks under eq. (1)  + actually implement them

– below eq. (12) do the authors mean A or A_eq?

– L168, please correct the highlighting

Author Response

We thank the Reviewer for the overall positive report on our manuscript. Concerning

below eq. (12) do the authors mean A or A_eq?

indeed we really mean A_eq. In fact the local free energy (in the presence of an external force) can be decomposed as 

A(x) = A_eq(x) - f\cdot x

and hence A_eq after Eq.12 is correct.  

Concerning 

A perhaps naive question to the Authors would be – have there been any MD or rather Brownian Dynamics simulations that would attempt to determine the transport properties of channels with deformed walls? I understand that in this case resolving the interactions, particularly hydrodynamic interactions, is a challenge but perhaps there have been numerical works focusing on addressing various aspects of this problem? I would appreciate if the Authors could comment on this. 

We have added a paragraph in which we address this point.

Round 2

Reviewer 1 Report

The authors have responded to my suggestions satisfactorily, substantially improving the writing. I do not have further comments.  The manuscript can be published as is.

Reviewer 3 Report

I am happy with the changes introduced by the Authors and I recommend the paper for publication.